# Prevalence of stroke in young adults in the Middle East and North Africa Region: A systematic review and meta-analysis

Salma Hegazi[1,2], Roaa Aly[1,2], Duaa Yousif[1,2], Salma Al-Nuaimi[1,2], Amgad Mohammed Elshoeibi[1,2], Elhassan Mahmoud[1,2], Abdalla Moustafa Elsayed[1,2], Tawanda Chivese[3]*

1 College of Medicine, QU Health, Qatar University, Doha, Qatar, 2 Hamad Medical Corporation, Doha, Qatar, 3 Department of Science and Mathematics, School of Interdisciplinary Arts and Sciences (SIAS), University of Washington Tacoma, Tacoma, Washington, United States of America

* tchivese@uw.edu

## Abstract

Young adults in the Middle East and North Africa (MENA) region are experiencing a high prevalence of risk factors for stroke, such as obesity, diabetes, hypertension and dyslipidemia. This systematic review and meta-analysis aimed to estimate the prevalence of stroke in this important demographic within the MENA region. Comprehensive searches of PubMed, Scopus, Embase, and CINAHL databases were conducted for observational studies published up to 2024. Eligible studies reported stroke prevalence in individuals aged 15–45 years within the MENA region. Prevalence estimates for young adults were pooled using the random effects model after using the Freeman-Tukey double-arcsine transformation. Heterogeneity was assessed using the $I^2$ statistic, and publication bias was evaluated using Doi plots. The risk of bias was assessed using the tool developed by Hoy. Overall, fourteen studies from Saudi Arabia, Iran, Egypt, Morocco, Lebanon, and Qatar, encompassing a total of 3,815,317 individuals, were included. The highest prevalence of stroke among young adults was reported in Saudi Arabia (0.167%, 95%CI 0.00–0.840), and the lowest prevalence was reported in Iran (0.011%, 95%CI 0.000–0.034). After meta-analysis, the overall prevalence of stroke among young adults was 0.036% (95%CI 0.024–0.051, $I^2$ = 96.9%), equivalent to 3.6 per 10,000. Stroke prevalence was comparable between females (0.038%, 95%CI 0.019–0.061, $I2$ = 94.8%) and males (0.037%, 95%CI 0.017–0.064, $I^2$ = 96.2%). The prevalence more than doubled from 0.022% (95%CI 0.009%-0.041%, $I^2$ = 96.2%) before 2015 to 0.072% (95%CI 0.034%-0.122%, $I2$ = 98.0%) after the year 2015. Ischemic stroke was the predominant subtype, accounting for 79.3% (95%CI 77.4–81.1) of the cases. While the prevalence of stroke in young adults in the MENA region is lower than in high-income regions, it appears to be increasing. There is a need for more robust data on stroke prevalence in the region.

**Data availability statement:** All the data used in the analysis are included in the manuscript and included materials.

**Funding:** The author(s) received no specific funding for this work.

**Competing interests:** The authors have declared that no competing interests exist.

## Registration

The protocol for this systematic review and meta-analysis was registered on the International Prospective Register of Systematic Reviews (PROSPERO) under the registration ID. CRD42024504772.

## 1. Introduction

The likelihood of having a stroke during one's lifetime has risen by 50% in the past 17 years, and currently, it is estimated that one in four individuals will encounter a stroke at some point in their lives [1]. Along with that high incidence, stroke ranks as the second most prevalent cause of mortality, and the third leading cause of disability [2,3]. Globally, the prevalence of stroke has been increasing, but with a higher burden in the low- and middle-income countries and the so-called developing countries [1,4]. These countries are undergoing demographic, nutritional and epidemiologic transitions that have raised the burden of cardiometabolic diseases, possibly contributing to an increasing prevalence of stroke [5,6]. The Middle East and North Africa (MENA) region, characterized by its distinctive socio-cultural and dynamic demographic shifts, is a region in which most countries are within the middle-income countries and developing countries grouping, despite some countries being classified as low or high income [7]. The prevalence of stroke has doubled during the period 1990–2021 in this region, necessitating the need for preventive strategies and more research in identifying the burden of stroke in key populations [8].

Historically, stroke was perceived as a disease of the elderly; however, it is now increasing in prevalence in young adults globally [9]. In the United States, stroke incidence among adults aged 20–44 has risen from 17 per 100,000 adults in 1993 to 28 per 100,000 adults in 2015 [10]. In Spain, the annual incidence of ischemic stroke among adults aged 18–50 was approximately 12.3 per 100,000 between 2005 and 2015 [11]. In several Western European populations, the incidence among those under 45 is around 10 per 100,000 [12]. Many studies define a young adult as either younger than or equal to 45 years old [13–15]. Young adults with ischemic stroke face a mortality risk up to four times greater than that of their healthy counterparts [9]. Moreover, the economic and emotional challenges posed by stroke are significant, with as many as one-third of individuals with stroke facing unemployment for a duration of up to eight years after the event [16]. This imposes a significant economic burden on societies, as the loss of young adults, who are often a critical component of the workforce, would result in decreased productivity and potential labour shortages in societies [17].

The prevalence of the major risk factors for stroke, such as obesity, diabetes, hypertension, and cardiovascular diseases, is increasing across all regions in young adults, especially in the MENA region [18]. The region has one of the highest prevalences of diabetes and cardiovascular diseases, making it a region of possible high prevalence of stroke [19,20]. These risk factors for stroke may differ in effect, magnitude and distribution in young adults compared to older adults [21]. However,

currently, the overall prevalence of and risk factors of stroke in young adults in the MENA region is not well quantified, as most studies report overall estimates across all age groups. This systematic review and meta-analysis aimed to estimate the prevalence of stroke in young adults in this region.

## 2. Methods

The conduct and reporting of this systematic review and meta-analysis adhered to the Preferred Reporting Items for Systematic Reviews and Meta-Analyses (PRISMA) guidelines [22]. The protocol for this study is registered on the International Prospective Register of Systematic Reviews (PROSPERO) under the registration ID CRD42024504772.

### 2.1. Information sources and search strategy

A comprehensive search of electronic databases, including PubMed, Scopus, Embase, and Cumulative Index to Nursing and Allied Health Literature (CINAHL), was conducted. Reference lists of included studies were also screened for additional relevant publications.

The search strategy combined controlled Medical Subject Heading (MeSH) terms and free-text keywords for stroke and its subtypes (e.g., ischemic stroke, hemorrhagic stroke) with regional terms (e.g., Middle East, North Africa, and individual MENA countries). The final database search was completed on the 29th of November 2024. The full details of the search terms and strategy for each database are provided in S1 Text.

### 2.2. Selection process

Two screeners independently carried out study selection through a three-step process. First, titles were reviewed to exclude irrelevant studies. Abstracts were then screened to identify potentially eligible studies. Finally, full-text articles were assessed for eligibility. A third reviewer resolved any disagreements between screeners.

### 2.3. Eligibility criteria

Screening of retrieved studies was performed by a team of reviewers, with independent double screening conducted by two reviewers. Discrepancies were resolved by a third reviewer. Observational studies published in the last 20 years (from 2004 onwards) were included if they reported stroke prevalence in young adults aged 15–45 years within the MENA region. This age range was selected based on commonly used definitions in stroke literature, where 45 years is often considered the upper limit of "young adult stroke" in global studies [13–15,23]. Studies were eligible if they reported prevalence data (total or subtype: ischemic or hemorrhagic), included age-disaggregated data for the target group (15–45), and provided extractable data specific to MENA populations. We included studies with narrower overlapping age brackets (e.g., 20–45) but excluded those with broader age groups (e.g., 15–60) if relevant age-specific data were not provided. Studies on comorbid or special populations were included if relevant. We excluded commentaries, conference abstracts, narrative reviews, and studies lacking sufficient outcome data or those whose results could not be converted into prevalence estimates. Although no language restrictions were applied during the search, all included studies were published in English or had English abstracts accessible through the databases searched.

### 2.4. Definition and assessment of outcomes

The primary outcome of the study was the total number of stroke cases. The accepted stroke definitions were the World Health Organization (WHO) guidelines, which describe stroke as a clinical syndrome with rapidly developing focal or global cerebral dysfunction lasting more than 24 hours or resulting in death, due to a vascular cause [24]. Other acceptable stroke definitions were such as those from the American Heart Association (AHA), which define stroke as a neurological deficit caused by an acute focal injury to the central nervous system due to disrupted blood supply, emphasizing the

 

role of imaging findings in the diagnosis [25]. Secondary outcomes assessed included the counts of stroke cases in males and females, the type of stroke (Ischemic and Hemorrhagic) and the trend over time of the number of stroke cases. All possible measures to diagnose stroke, including clinical criteria and imaging, were included.

### 2.5. Data extraction process

Data extraction was performed by two reviewers independently using a pre-defined form on Microsoft Excel. Extracted data included the study citation, years of data collection, region, country, study design, sample size, total number of individuals with stroke, age groups, subtype of stroke, and the numbers of stroke cases in males and females.

### 2.6. Study risk of bias assessment

The quality of the included studies was evaluated by independent reviewers using the Hoy risk of bias tool [26]. The tool comprises ten domains that provide an evaluation of both internal and external validity in addition to a summary risk of bias judgement. Each domain was rated as "Yes" (low risk) or "No" (high risk), and total scores ranged from 0 (high risk of bias) to 10 (low risk). A score of 8 or more was interpreted as low risk, consistent with published guidelines.

The assessment of internal validity considers several factors, such as whether data were collected directly from the subjects rather than through proxies and if an acceptable case definition was used in the study. The tool also evaluates whether the study instrument was demonstrated to be both valid and reliable for measuring the parameter of interest and whether the same mode of data collection was applied consistently across all subjects. The duration of the shortest prevalence period for the parameter of interest is also reviewed to ensure its appropriateness, as well as the accuracy in defining the numerator(s) and denominator(s) for the parameter. For external validity, the assessment examines whether the study's target population closely resembled the national population in relevant variables and whether the sampling frame was an accurate representation of the target population. The tool also considers whether random selection or a census was used in the sample selection and whether steps were taken to minimize potential nonresponse bias.

### 2.7. Synthesis methods

All statistical analysis was conducted on Stata 17.0 (College Station, Texas, USA). A meta-analysis was conducted, and prevalence estimates were pooled using a random-effects model, after using the Freeman-Tukey transformation to stabilise variances of prevalence estimates [27]. Sensitivity analyses using leave-one-out were performed to assess the robustness of the results. Heterogeneity was assessed using the $I^2$ statistic and Cochran's Q test, with $I^2$ levels up to 25%, 50%, and 75% indicating low, moderate, and high levels of heterogeneity, respectively. Subgroup analysis included analysis by subregion (North Africa vs. Middle East), analysis by sex, type of stroke, and time period when the study was done. Publication bias was evaluated using funnel plots and Doi plots with the LFK index [28]. LFK index values greater than 1 or less than -1 indicate minor asymmetry, while values greater than 2 or less than -2 indicate major asymmetry.

## 3. Results

### 3.1. Search results

Fig 1 provides an overview of the study search and selection process. A total of 6331 records were identified from database searches across Embase (2498 records), CINAHL (441 records), Scopus (1753 records), and PubMed (1639 records). After removing 3379 duplicate records, 2952 records were screened for eligibility. During the screening process, 2821 records were excluded based on title and abstract screening. A total of 131 reports were sought for full-text review, of which 117 records were excluded for the following reasons: nine lacked full text, 44 were conference abstracts, three were letters to the editor, and 61 did not report relevant outcomes of interest. Finally, 14 studies met the inclusion criteria and were included.

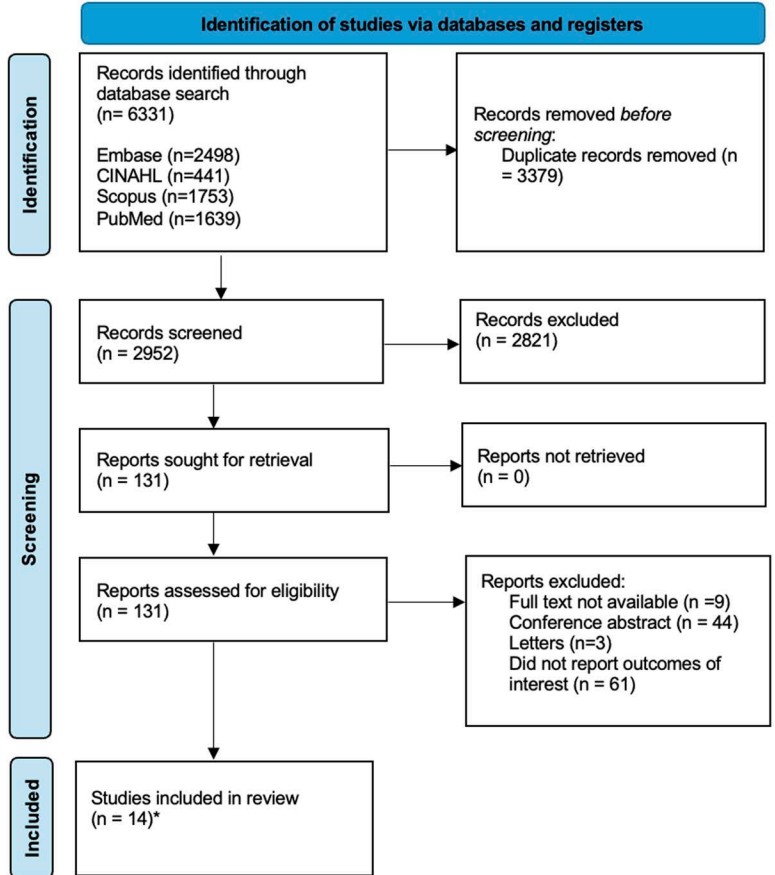

**Fig 1. PRISMA Flow Diagram of Study Selection Process.** *Two studies were excluded from the overall prevalence meta-analysis due to missing total prevalence data, but were included in subtype analyses.

## 3.2. Characteristics of included studies

Table 1 shows the characteristics of the 14 studies included in this meta-analysis. The study by Azarpazhooh et al. [29] contained two independent populations that were analyzed separately in the meta-analysis. In Table 1, these populations are labelled as "Azarpazhooh et al. (a)" for the first population and "Azarpazhooh et al. (b)" for the second population. Although the total number of included studies was 14, the inclusion of these independent populations brought the number of assessed population reports in the meta-analysis to 15 studies.

The included studies were from Egypt, Iran, Saudi Arabia, Qatar, Morocco, and Lebanon. Egypt contributed the most studies (n = 4), followed by Iran (n = 3). Saudi Arabia, Qatar, and Morocco each contributed two studies, while Lebanon provided one study. The studies were conducted between 2006 and 2022, encompassing a total of 2424 individuals, under the age of 45 years, with stroke. The studies comprised seven cross-sectional studies, five prospective observational studies, and two retrospective studies. Stroke definitions varied, incorporating WHO criteria, Saudi Ministry of Health guidelines, and clinical criteria. The diagnosis was based only on imaging and clinical examination in 11 studies. The remaining studies used additional tools like questionnaires to determine stroke status.

**Table 1. Characteristics of Included Studies.**

| Author | Year | Country | Study Type | Setting | Age Range Included | Sample Size | Number of Individuals with Stroke | Male Distribution | Stroke Definition | Diagnostic Criteria | Data Collection |
|---|---|---|---|---|---|---|---|---|---|---|---|
| Azarpazhooh et al. (a) [29] | 2013 | Iran | Prospective cohort study | Population-based | 15-44 | 24021 | 2 | 0% | WHO criteria | Clinical exam and imaging | Pilgrim medical history and evaluation |
| Azarpazhooh et al. (b) [29] | 2013 | Iran | Mashhad Stroke Incidence Study (MSIS) | Population-based | 15-44 | 253462 | 3 | 33% | WHO criteria | Clinical exam and imaging | Pilgrim medical history and evaluation |
| Abujaber et al. [30] | 2024 | Qatar | Retrospective analytical study | Hospital-based | ≤40 | – | 1746 | – | Clinical diagnosis | Clinical exam and imaging | The hospital stroke registry |
| Alhazzani et al. [31] | 2018 | Saudi Arabia | Prospective observational study | Hospital-based | <40-44 | 1710560 | 187 | 59% | Saudi Ministry of Health Guidelines | Clinical exam and imaging | Stroke patient data from hospitals |
| Chraa et al. [32] | 2014 | Morocco | Retrospective cohort study | Population-based | 18-45 | 1000000 | 128 | 59% | Clinical diagnosis | Clinical exam and imaging | Retrospective analysis of patient records |
| Khedr et al. [33] | 2014 | Egypt | Cross sectional study | Population-based | <20-39 | 6018 | 4 | 50% | Clinical diagnosis | Questionnaire, clinical exam, imaging, and lab tests | Three-phase random door-to-door study |
| Lahoud et al. [34] | 2015 | Lebanon | Cross sectional study | Population-based | <40 | 5213 | 3 | 0% | WHO criteria | Questionnaire and clinical exam | Multi-stage sampling via telephone interviews |
| Al-Rubeaan et al. [35] | 2016 | Saudi Arabia | Cross-sectional study | Population (Diabetic people)-based | 25-44 | 11197 | 57 | 67% | Clinical diagnosis | Clinical exam and imaging | Data from Saudi National Diabetes Registry |
| Amiri et al. [36] | 2018 | Iran | Cohort study | Population-based | <45 | 369343 | 58 | 45% | NR | Clinical exam and imaging | Multiple sources used for stroke data |
| El Tallawy et al. [37] | 2015 | Egypt | Cross-sectional study | Population-based | 20-<40 | 31998 | 21 | 38% | Clinical diagnosis | Questionnaire, clinical exam, imaging, and lab tests | Door-to-door stroke screening study |
| El Tallawy et al. [38] | 2013 | Egypt | Cross-sectional study | Population-based | 20-<40 | 11664 | 3 | – | WHO criteria | Clinical exam and imaging | Door-to-door survey and evaluations |
| Engels et al. [39] | 2014 | Morocco | Cross-sectional study | Population-based | 15-44 | 30740 | 8 | – | WHO criteria | Clinical exam and imaging | Household survey and neurologist confirmation |
| Farghaly et al. [40] | 2013 | Egypt | Cross-sectional study | Population-based | <20-<40 | 47101 | 25 | – | WHO criteria | Clinical exam and imaging | Door-to-door survey with neurological evaluation |
| Ghandehari et al. [41] | 2006 | Iran | Prospective observational study | Population-based | 15-45 | 314000 | 124 | 52% | Clinical diagnosis | Clinical exam, blood tests and imaging | Prospective data from hospitalized patients |
| Khan et al. [42] | 2008 | Qatar | Prospective observational study | Hospital-based | 15-45 | – | 55 | 78% | WHO criteria | Clinical exam and imaging | Comprehensive identification and examination process |

### 3.3. Risk of bias in studies

The 14 included studies demonstrated a wide range in methodological quality. Eight studies [33,34,36–41] scored 8 out of 10, indicating a low risk of bias and strong adherence to internal and external validity safeguards. The study by Al-Rubeaan [35] scored 7, while the study by Azarpazhooh [29] scored 6, and four studies [30–32,42] received scores of

4, reflecting a higher risk of bias. The most frequent sources of bias were related to non-representative sampling frames, lack of national representativeness in the target population, and insufficient information on the reliability or validity of the measurement instruments used to assess stroke prevalence. These limitations were particularly evident in studies that relied on hospital-based convenience samples or lacked methodological transparency. Nevertheless, most studies used consistent case definitions and data collection procedures, supporting the overall robustness of the findings. Detailed item-by-item scoring for each study is provided in S1 Table.

## 3.4. Stroke prevalence results

**3.4.1. Stroke prevalence.** Fig 2 shows the twelve studies [29,31–41] that were included in the final analysis of stroke prevalence among individuals younger than 45 years, encompassing a combined sample size of 3,815,317 individuals. Two studies [30,42] were excluded from this analysis due to missing total prevalence data, but were included in the ischemic and hemorrhagic subtype analyses. The prevalence of stroke reported in the included studies ranged from 0.001% in a study from Iran [29] to 0.509% in a study from Saudi Arabia [35]. After meta-analysis, the overall synthesized prevalence of stroke in this population was 0.036% (95%CI 0.024–0.051), with high heterogeneity ($I^2 = 96.9\%$, P < 0.0001)

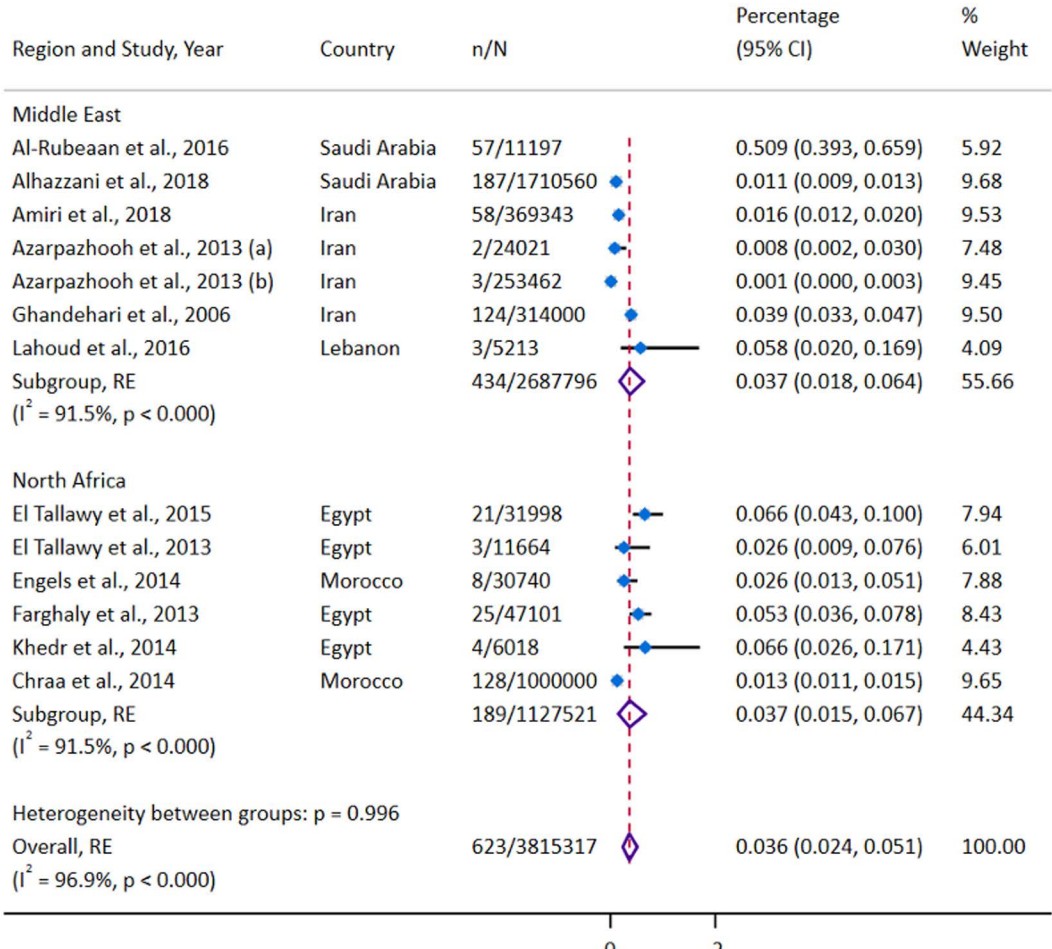

**Fig 2. Forest plot of stroke prevalence among individuals younger than 45 years by Regions (Middle East and North Africa).**

(Fig 2). This overall prevalence equates to a rate of 3.6 cases of stroke per 10,000 of the population of young adults. A leave-one-out sensitivity analysis revealed that the study by Al-Rubeaan et al. [35] had a more significant impact on the overall effect size when compared to other studies (S1 Fig). When the study by Al-Rubeaan et al. [35] was excluded, the pooled prevalence decreased to 0.023% (95%CI 0.015–0.033). Assessment of publication bias using funnel and Doi plots (S2, S3 Figs) showed major positive asymmetry (LFK=4.38), suggesting that smaller studies were disproportionately reporting higher prevalence estimates. Subgroup analysis by region revealed that the prevalence of stroke was the same in the Middle East and North Africa (Fig 2). When analyzed by country, the prevalence was highest in Saudi Arabia (0.167%, 95%CI 0.00–0.840) and lowest in Iran (0.011%, 95%CI 0.000–0.034). Subgrouping by the type of data used for prevalence estimation showed that estimates from hospital-based studies (0.059%, 95%CI 0.030–0.099) were higher than those from population-based studies (0.029%, 95%CI 0.014–0.050).

**3.4.2. Stroke prevalence by sex.** A total of eight studies [29,31,33–37,41] analyzed stroke prevalence among males and females younger than 45 years, encompassing 1,278,428 females and 1,427,717 males (Figs 3 and 4). The overall pooled prevalence of stroke was 0.038% (95%CI 0.019–0.061) in females and 0.037% (95%CI 0.017–0.064) in males, with high heterogeneity for both groups (females: $I^2$=94.8%, P<0.0001; males: $I^2$=96.2%, P<0.000). A leave-one-out sensitivity analysis identified the study by Al-Rubeaan [35] as an influential study in both groups (S4, S5 Figs). Excluding this study reduced the pooled prevalence to 0.024% (95%CI 0.012–0.042) for females and 0.016% (95%CI 0.006–0.030) for males. Publication bias was evident for analyses for both sexes, with funnel and Doi plots (S6–S9 Figs) showing major positive asymmetry (females: LFK=5.90; males: LFK=3.11), indicating smaller studies disproportionately reported a higher prevalence of stroke in the young.

**3.4.3. Prevalence of ischemic and hemorrhagic stroke subtypes.** A total of four studies (S10, S11 Figs) [30,37,40,42] with 1,847 individuals younger than 45 years reported data on the type of stroke. Among these studies, the prevalence of ischemic stroke within the total of stroke cases ranged between 72.7% [42] and 81% [37]. The overall synthesized prevalence of ischemic stroke among young adults with stroke was 79.3% (95%CI 77.4 - 81.1, $I^2$=0.0%). For hemorrhagic stroke, prevalence ranged from 19% [37] to 27.3% [42]. The overall pooled prevalence of hemorrhagic stroke among young adults with stroke was 21.7% (95%CI 19.8-23.6). The funnel and Doi plots (S12–S15 Fig) showed evidence of major asymmetry, suggesting that smaller studies were disproportionately reporting higher prevalence rates.

| Study, Year | Country | n/N | Percentage (95% CI) | % Weight |
|---|---|---|---|---|
| Al-Rubeaan et al., 2016 | Saudi Arabia | 38/4687 | 0.811 (0.591, 1.111) | 8.05 |
| Alhazzani et al., 2018 | Saudi Arabia | 110/920598 | 0.012 (0.010, 0.014) | 14.52 |
| Amiri et al., 2018 | Iran | 26/186087 | 0.014 (0.010, 0.020) | 14.29 |
| Azarpazhooh et al., 2013 (a) | Iran | 0/9639 | 0.000 (0.000, 0.040) | 10.46 |
| Azarpazhooh et al., 2013 (b) | Iran | 1/126926 | 0.001 (0.000, 0.004) | 14.16 |
| El Tallawy et al., 2015 | Egypt | 8/15760 | 0.051 (0.026, 0.100) | 11.75 |
| Ghandehari et al., 2006 | Iran | 64/158256 | 0.040 (0.032, 0.052) | 14.24 |
| Khedr et al., 2014 | Egypt | 2/3171 | 0.063 (0.017, 0.230) | 6.63 |
| Lahoud et al., 2016 | Lebanon | 0/2593 | 0.000 (0.000, 0.148) | 5.91 |
| Overall, RE ($I^2$ = 96.2%, p < 0.000) | | 249/1427717 | 0.037 (0.017, 0.064) | 100.00 |

**Fig 3. Forest Plot of Stroke Prevalence Among Males Younger Than 45 Years.**

| Study, Year | Country | n/N | | Percentage (95% CI) | % Weight |
|---|---|---|---|---|---|
| Al-Rubeaan et al., 2016 | Saudi Arabia | 19/6510 | | 0.292 (0.187, 0.455) | 8.40 |
| Alhazzani et al., 2018 | Saudi Arabia | 77/770295 | | 0.010 (0.008, 0.012) | 14.96 |
| Amiri et al., 2018 | Iran | 32/183256 | | 0.017 (0.012, 0.025) | 14.65 |
| Azarpazhooh et al., 2013 (a) | Iran | 2/14382 | | 0.014 (0.004, 0.051) | 11.09 |
| Azarpazhooh et al., 2013 (b) | Iran | 2/126536 | | 0.002 (0.000, 0.006) | 14.47 |
| El Tallawy et al., 2015 | Egypt | 13/16238 | | 0.080 (0.047, 0.137) | 11.43 |
| Ghandehari et al., 2006 | Iran | 60/155744 | | 0.039 (0.030, 0.050) | 14.57 |
| Khedr et al., 2014 | Egypt | 2/2847 | | 0.070 (0.019, 0.256) | 5.36 |
| Lahoud et al., 2016 | Lebanon | 3/2620 | | 0.115 (0.039, 0.336) | 5.08 |
| Overall, RE ($I^2$ = 94.8%, p < 0.000) | | 210/1278428 | | 0.038 (0.019, 0.061) | 100.00 |

0        .2

**Fig 4. Forest Plot of Stroke Prevalence Among Females Younger Than 45 Years.**

**3.4.4. Trends in the prevalence of stroke in the young over time.** To assess trends in the prevalence of stroke in the young over time, studies were divided into two groups: those conducted before 2015 and those conducted on or after 2015 (S16 Fig). Among the 8 studies [29,32,33,38–42] conducted before 2015, a total of 1,686,006 individuals were assessed, with 352 reported stroke events. In contrast, the 6 studies [30,31,34–37] conducted in 2015 or later included a larger combined sample of 2,128,311 individuals and reported 326 stroke events. The pooled prevalence estimate for studies conducted on or after 2015 was 0.072% (95%CI 0.034%-0.122%), with high heterogeneity ($I^2$ = 98.0%). In contrast, studies conducted before 2015 reported a significantly lower pooled prevalence of 0.022% (95%CI 0.009%-0.041%), also with high heterogeneity (I2 = 96.2%). The p for interaction (p = 0.014) indicates a statistically significant difference in pooled prevalence estimates between the two periods; however, this should be interpreted with caution due to the small number of studies and high heterogeneity.

## 4. Discussion

In this systematic review and meta-analysis of 14 studies, the overall pooled prevalence of stroke in young adults in the MENA region was estimated at 0.036%, with variations by country and sex. Further, our findings showed that ischemic stroke contributed the biggest proportion (79%) of stroke in young adults in the MENA region. Moreover, there is an observed increase in stroke prevalence in studies conducted on or after 2015 compared to earlier studies, suggesting a potential rising trend over time, although differences in study methodologies might explain this variation.

In our current analysis, the overall prevalence of stroke among young adults in the MENA region was 0.036% (3.6 per 10,000) with nearly equal prevalence between males (0.037%) and females (0.038%), which appears lower than the global estimates. For instance, In Sweden, stroke prevalence in individuals under 45 years ranged from 6.4 to 7.6 per 10,000 over a 15-year period [43], while in the U.S., hospitalization rates for acute ischemic stroke were significantly higher, particularly among young males (11.2 to 18.0 per 10,000 for ages 18–34 and 37.7 to 68.2 per 10,000 for ages 35–44) and females (3.8 to 5.8 per 10,000 for ages 18–34 and 24.8 to 35.8 per 10,000 for ages 35–44) [44]. Although stroke prevalence in the MENA region remains lower than other global estimates, our analysis suggests a rising trend over time, with studies conducted on or after 2015 reporting a higher pooled prevalence (0.072%) compared to those before 2015 (0.022%). These findings suggest an increase in the number of affected young adults over a short period of time. The findings also underscore the need to address the risk factors for stroke, enhanced screening, and consider allocations

of more resources for the treatment of stroke and rehabilitation for young adults in the region. However, this should not be interpreted as a confirmed increase in stroke prevalence over time. Additional high-quality, population-based studies with clearly defined time frames are necessary to accurately assess temporal changes.

We found that the prevalence of stroke in young adults was highest in Saudi Arabia (0.167%) and lowest in Iran (0.011%). However, data from several MENA countries remain unavailable, highlighting a need for more comprehensive epidemiological studies in the region. Our analysis also revealed no significant difference in stroke prevalence between males (0.037%) and females (0.038%), suggesting that sex may not be a major determinant of stroke burden in young adults. In terms of stroke subtypes, ischemic stroke was the most common, accounting for 79.3% of cases, a trend consistent with global data indicating that ischemic strokes are more prevalent in young adults than hemorrhagic strokes [2,42].

Several studies have identified the key risk factors that contribute to stroke in young adults, including hypertension, diabetes, smoking, and obesity [45–48]. These risk factors are well-established in older populations, but their impact on younger adults is less frequently studied. The high prevalence of hypertension and diabetes in the MENA region, alongside rising rates of smoking and obesity, may explain the increasing incidence of stroke in this age group. These findings are consistent with the global trend of increasing cardiometabolic diseases in young adults, particularly in low- and middle-income countries [4]. Hypertension is highly prevalent in the MENA region [49], in addition to being one of the commonly reported risk factors for stroke across the studies. Additionally, hypertension is a leading modifiable risk factor for stroke in both young and older populations [49]. The increasing rates of hypertension and other cardiovascular risk factors in the MENA region are likely driven by dietary patterns, sedentary lifestyles, and limited access to healthcare in some areas. Similarly, the high prevalence of diabetes in MENA countries, which ranks among the highest globally, is a major contributor to stroke risk in this demographic. The combination of obesity and metabolic dysfunction exacerbates the risk of stroke, making the prevention and management of these conditions critical in reducing stroke incidence in young adults.

Our study has several limitations. First, although we aimed to cover the MENA region comprehensively, all the 14 included studies were from only six countries, namely, Egypt, Iran, Saudi Arabia, Morocco, Qatar, and Lebanon. Thus, the findings may not be generalizable to all MENA countries. Second, there was considerable heterogeneity across included studies in terms of age ranges, study design (cross-sectional, retrospective, or prospective), diagnostic methods (clinical vs. imaging-based diagnosis), screening approach, hospital vs population-based and stroke definitions (WHO vs. local or unspecified criteria). These differences may affect pooled prevalence estimates and limit direct comparability. In particular, hospital-based studies may overestimate stroke prevalence due to selection bias and unclear or unrepresentative denominators. Although we conducted subgroup analysis by study setting (population- vs. hospital-based), some uncertainty remains. However, given the overall scarcity of data in the region, these studies were included to ensure a more comprehensive and representative synthesis of the available evidence.

The analysis showing an increase in stroke prevalence after 2015 should be interpreted with caution. The grouping was based on publication year rather than year of data collection, which may introduce misclassification. The year was selected as it represented the midpoint of the publication timeline, allowing for an evaluation of trends in stroke prevalence over time. While it is unlikely that any specific causative factor emerged in that particular year, the observed doubling in prevalence likely reflects a broader, gradual increase across the studied period.

Finally, smaller studies appeared to report higher prevalence estimates [45,46]. This may reflect publication bias or small study effects, a known trend in meta-analyses where small studies tend to overestimate effect sizes. Although we used funnel and Doi plots to detect this and conducted sensitivity analyses, we recommend cautious interpretation of these estimates.

This study has several strengths. It is, to our knowledge, the most comprehensive systematic review and meta-analysis on stroke prevalence and risk factors in young adults in the MENA region. By including data from multiple countries, the study provides a broad overview of the epidemiological landscape of stroke in young adults across this diverse region. Moreover, the study highlights critical gaps in knowledge and provides valuable evidence to inform

future research and public health strategies. Unlike prior Global Burden of Disease (GBD) studies such as those by Shahbandi and Jaberinezhad [47,48], which provide modeled estimates across all ages, our study offers a focused meta-analysis of observational data specific to young adults aged 15–45 years in the MENA region. By using real-world prevalence data, we provide detailed insights on stroke subtypes, sex differences, and study settings, granularity not available in GBD outputs. Compared to GBD estimates, our findings suggest a lower absolute prevalence in young adults (15–45 years). The GBD 2021 data indicate that 14.8% of all stroke cases occur in the 15–49 age group globally [49], and reports age-standardized stroke prevalence across all ages in the MENA region was approximately 607 per 100,000 [48]. In our study, the pooled prevalence in 15–45-year-olds was 36 per 100,000, but it should be noted that our meta-analytic estimate is not comparable to an age-standardized estimate. The age-standardized prevalence from the GBD is for comparative purposes and is highly dependent on the standardizing population used. Additionally, the GBD suggests that the age-standardized prevalence of stroke in the MENA region went down from 744 per 100,000 in 1990–607 per 100,000 in 2021, a trend that could not be assessed in the current meta-analysis because of a lack of data sources. Our findings, together with the GBD estimates, highlight a need for more robust data on stroke in young people in the MENA region.

## 5. Conclusion

This systematic review and meta-analysis revealed an overall stroke prevalence of 0.036% among young adults in the MENA region, with ischemic stroke being the predominant subtype and significant heterogeneity across countries. Gender-specific analyses showed comparable prevalence in males and females, although certain countries exhibited higher rates among women. Our analysis also suggested that the prevalence of stroke in young adults in the region may be increasing, although the data sources were few and mostly of low quality. More robust, standardized, and representative epidemiological studies are needed to accurately assess the burden of stroke and its trends among young adults in the MENA region.

## Supporting information

**S1 Text. Search Strategy This document shows the search strategy that was used to identify eligible studies from databases.**
(DOCX)

**S1 PRISMA Checklist. PRISMA Checklist.**
(PDF)

**S1 Table. Risk of Bias Assessment - Detailed Hoy Scale Scores for Each Included Study (Adapted from Hoy et al. 2012 [1]).**
(DOCX)

**S2 Table. Extracted Data from the included studies.**
(DOCX)

**S3 Table. Studies Excluded from the Systematic Review and Reasons for Exclusion.**
(DOCX)

**S1 Fig. Influence Analysis of Stroke Prevalence in the MENA Region Among Individuals Younger Than 45 Years.**
(TIF)

**S2 Fig. Funnel Plot of Stroke Prevalence in the MENA Region Among Individuals Younger Than 45 Years.**
(TIF)

**S3 Fig. Doi Plot of Stroke Prevalence in the MENA Region Among Individuals Younger Than 45 Years.**
(TIF)

**S4 Fig. Influence Analysis of Male Stroke Prevalence in the MENA Region Among Individuals Younger Than 45 Years.**
(TIF)

**S5 Fig. Influence Analysis of Female Stroke Prevalence in the MENA Region Among Individuals Younger Than 45 Years.**
(TIF)

**S6 Fig. Funnel Plot of Male Stroke Prevalence in the MENA Region Among Individuals Younger Than 45 Years.**
(TIF)

**S7 Fig. Doi Plot of Male Stroke Prevalence in the MENA Region Among Individuals Younger Than 45 Years.**
(TIF)

**S8 Fig. Funnel Plot of Female Stroke Prevalence in the MENA Region Among Individuals Younger Than 45 Years.**
(TIF)

**S9 Fig. Doi Plot of Female Stroke Prevalence in the MENA Region Among Individuals Younger Than 45 Years.**
(TIF)

**S10 Fig. Forest Plot of Ischemic Stroke Prevalence in the MENA Region Among Individuals Younger Than 45 Years.**
(TIF)

**S11 Fig. Forest Plot of Hemorrhagic Stroke Prevalence in the MENA Region Among Individuals Younger Than 45 Years.**
(TIF)

**S12 Fig. Funnel Plot of Ischemic Stroke Prevalence in the MENA Region Among Individuals Younger Than 45 Years.**
(TIF)

**S13 Fig. Doi Plot of Ischemic Stroke Prevalence in the MENA Region Among Individuals Younger Than 45 Years.**
(TIF)

**S14 Fig. Funnel Plot of Hemorrhagic Stroke Prevalence in the MENA Region Among Individuals Younger Than 45 Years.**
(TIF)

**S15 Fig. Doi Plot of Hemorrhagic Stroke Prevalence in the MENA Region Among Individuals Younger Than 45 Years.**
(TIF)

**S16 Fig. Forest Plot of Stroke Prevalence Among Individuals Younger Than 45 Years by Year.**
(TIF)

## Author contributions

**Conceptualization:** Salma Hegazi, Salma Al-Nuaimi, Tawanda Chivese.

**Data curation:** Salma Hegazi, Roaa Aly, Duaa Yousif, Salma Al-Nuaimi, Amgad Mohammed Elshoeibi, Elhassan Mahmoud, Abdalla Moustafa Elsayed, Tawanda Chivese.

**Formal analysis:** Salma Hegazi, Amgad Mohammed Elshoeibi, Elhassan Mahmoud, Tawanda Chivese.

**Funding acquisition:** Tawanda Chivese.

**Investigation:** Salma Hegazi, Roaa Aly, Duaa Yousif, Salma Al-Nuaimi, Amgad Mohammed Elshoeibi, Elhassan Mahmoud, Tawanda Chivese.

**Methodology:** Salma Hegazi, Roaa Aly, Duaa Yousif, Salma Al-Nuaimi, Abdalla Moustafa Elsayed, Tawanda Chivese.

**Project administration:** Salma Hegazi, Tawanda Chivese.

**Resources:** Salma Hegazi, Roaa Aly, Duaa Yousif, Salma Al-Nuaimi, Amgad Mohammed Elshoeibi, Elhassan Mahmoud, Abdalla Moustafa Elsayed, Tawanda Chivese.

**Software:** Salma Hegazi, Amgad Mohammed Elshoeibi, Elhassan Mahmoud, Tawanda Chivese.

**Supervision:** Salma Hegazi, Tawanda Chivese.

**Validation:** Salma Hegazi, Roaa Aly, Duaa Yousif, Salma Al-Nuaimi, Tawanda Chivese.

**Visualization:** Salma Hegazi, Roaa Aly, Duaa Yousif, Salma Al-Nuaimi, Amgad Mohammed Elshoeibi, Elhassan Mahmoud, Abdalla Moustafa Elsayed, Tawanda Chivese.

**Writing – original draft:** Salma Hegazi, Roaa Aly, Duaa Yousif, Salma Al-Nuaimi, Amgad Mohammed Elshoeibi, Elhassan Mahmoud, Abdalla Moustafa Elsayed, Tawanda Chivese.

**Writing – review & editing:** Salma Hegazi, Tawanda Chivese.

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
