## [Decision Letter · Decision Letter 0]

17 Jun 2025

PGPH-D-25-01063

Prevalence of Stroke in Young Adults in the Middle East and North Africa Region: A Systematic Review and Meta-Analysis

Dear Dr. Chivese,

Thank you for submitting your manuscript to PLOS Global Public Health. After careful consideration, we feel that it has merit but does not fully meet PLOS Global Public Health’s publication criteria as it currently stands. Therefore, we invite you to submit a revised version of the manuscript that addresses the points raised during the review process.

We look forward to receiving your revised manuscript.

Kind regards,

Feten Fekih-Romdhane

Academic Editor

Journal Requirements:

1. As required by our policy on Data Availability, please ensure your manuscript or supplementary information includes the following:

2. We note that your Data Availability Statement is currently as follows: [All the data used in the analysis are included in the manuscript and included materials]

Additional Editor Comments (if provided):

Reviewers' comments:

Reviewer's Responses to Questions

**Comments to the Author**

1. Does this manuscript meet PLOS Global Public Health’s publication criteria?

Reviewer #1: Yes

Reviewer #2: Yes

Reviewer #3: Partly

2. Has the statistical analysis been performed appropriately and rigorously?

Reviewer #1: Yes

Reviewer #2: N/A

Reviewer #3: Yes

3. Have the authors made all data underlying the findings in their manuscript fully available (please refer to the Data Availability Statement at the start of the manuscript PDF file)?

Reviewer #1: Yes

Reviewer #2: No

Reviewer #3: Yes

4. Is the manuscript presented in an intelligible fashion and written in standard English?

Reviewer #1: Yes

Reviewer #2: Yes

Reviewer #3: Yes

Reviewer #1: This paper is written well and very clear to understand stroke prevalence among young adult. However, I think the introduction can be more specify to show number of stroke rate among young adult from other region for comparision with MENA region.

Reviewer #2: Prevalence of Stroke in Young Adults in the Middle East and North Africa Region: A Systematic Review and Meta-Analysis

The manuscript had performed a systematic review and meta-analysis that capable to estimate the stroke prevalence in MENA region. It well written and organized.

Major comments:

1. Why ages less than 15 and ages greater than 45 are excluded? Which standard age classification is employed to select ages between 15-45 to define young adults?

2. The 14 studies included in the meta –analysis originate from 6 countries i.e. Saudi Arabia, Iran, Egypt, Morocco, Lebanon, and Qatar> Is it logical/epidemiological representative to assume the overall prevalence computed from these applies to the whole MENA region?

It is notable that the region consists of the following countries……. This region includes Afghanistan, Algeria, Bahrain, Egypt, Iran (Islamic Republic of), Iraq, Jordan, Kuwait, Lebanon, Libya, Morocco, Oman, Palestine, Qatar, Saudi Arabia, Sudan, Syrian Arab Republic (SAR), Tunisia, Turkey, United Arab Emirates (UAE), and Yemen.

3. What is the causative agent for stroke prevalence doubling in the year 2015? It should be discussed and interpreted in the manuscript.

4. Excel extracted data can be included as supplementary data in PLOS as open publication mechanism…

5. In the methodology it is stated that publications in all languages were included but the references did not pin point any additions than English language publications. The region being mainly Arabic speaker…….Is there any attempt to assess Arabic based national or regional publications on the subject?

6. The study is claimed to be a new and Nobel. What values it adds on previous global burden of disease studies that includes the following manuscripts and others not mentioned here?

Shahbandi A, Shobeiri P, Azadnajafabad S, Saeedi Moghaddam S, Sharifnejad Tehrani Y, Ebrahimi N, Rezaei N, Rashidi MM, Ghamari SH, Abbasi-Kangevari M, Koolaji S, Haghshenas R, Rezaei N, Larijani B, Farzadfar F. Burden of stroke in North Africa and Middle East, 1990 to 2019: a systematic analysis for the global burden of disease study 2019. BMC Neurol. 2022 Jul 27;22(1):279. doi: 10.1186/s12883-022-02793-0. PMID: 35896999; PMCID: PMC9327376.

Jaberinezhad, M., Farhoudi, M., Nejadghaderi, S.A. et al. The burden of stroke and its attributable risk factors in the Middle East and North Africa region, 1990–2019. Sci Rep 12, 2700 (2022). https://doi.org/10.1038/s41598-022-06418-x

Al-Rukn S, Mazya M, Akhtar N, et al. Stroke in the Middle-East and North Africa: A 2-year prospective observational study of intravenous thrombolysis treatment in the region. Results from the SITS-MENA Registry. International Journal of Stroke. 2019;15(9):980-987. doi:10.1177/1747493019874729

Soleimani H, Nasrollahizadeh A, Nasrollahizadeh A, Razeghian I, Molaei MM, Hakim D, Nasir K, Al-Kindi S, Hosseini K. Cardiovascular disease burden in the North Africa and Middle East region: an analysis of the global burden of disease study 1990-2021. BMC Cardiovasc Disord. 2024 Dec 19;24(1):712. doi: 10.1186/s12872-024-04390-0. PMID: 39702074;

Mansouri A, Khosravi A, Mehrabani-Zeinabad K, Kopec JA, Adawi KI, Lui M, Rahim HF, Anwar W, Fadhil I, Sulaiman K, Bazargani N. Trends in the burden and determinants of hypertensive heart disease in the Eastern Mediterranean region, 1990–2019: an analysis of the Global Burden of Disease Study 2019. EClinicalMedicine. 2023 Jun 1;60

Vollset SE, Ababneh HS, Abate YH, Abbafati C, Abbasgholizadeh R, Abbasian M, Abbastabar H, Abd Al Magied AH, Abd ElHafeez S, Abdelkader A, Abdelmasseh M. Burden of disease scenarios for 204 countries and territories, 2022–2050: a forecasting analysis for the Global Burden of Disease Study 2021. The Lancet. 2024 May 18;403(10440):2204-56.

Minor:

There are variations in the methodologies employed for the 14 studies included in the study with regard to Stroke Definition Diagnostic Criteria Data Collection as shown on Table 1. This in turn might result in different interpretations and it should be emphasized in the discussions and limitations of the study adequately.

Regarding statistical techniques employed it is better if a subject expert on statistics ( Biostatistician) evaluates them.

In conclusion, the study had some unique features worthy of publications in PLOS Global Public health. The authors had performed great piece of work.

Reviewer #3: This systematic review looks at the prevalence of stroke in young adults in The Middle East and North Africa. While there is a lot of work that has been put into this, some methodological aspects need to be expanded.

Methods:

Can you please expand a bit on your exclusion criteria. For example:

1. What did you do if a study included people with a larger age range than 15-45 (for example 15-60) did you include it, or exclude it?

2. What did you do if a study included people with a smaller age range than 14-45 (for example 20-45) did you include it or exclude it?

3. How did you deal with studies that had data from within MENA and outside MENA?

4. How did you deal with special populations?

5. Did you include studies that were based on people with multi-morbidities?

For risk of bias, many of your readers are probably unfamiliar with Hoy-can you clarify if 10 is the best possible score?

Results:

I see that you included several studies that were hospital based. Hospital based studies can be problematic in conducting prevalence studies as it can be challenging to understand the denominator that the population is coming from-use of hospital based studies generally inflates prevalence estimates. I would like to see some mention of how you dealt with this, and how you estimated the denominators.

I was a bit surprised to see that all of your studies were a low risk of bias. You may want to go through those again. I have done a fair number of systematic reviews and have never done one where every single study was low risk of bias.

At the beginning of section 3.4.1 you state that 12 studies were included for meta-analysis, but do not explain why some were excluded. In your results you need to have a section were you explain this. You may also want to include this in your prisma diagram.

In section 3.4.4 when you discuss trends over time, it would be helpful to also discuss the number of studies in each time frame, the number of individuals counted, and the number of events. Given the small number of studies in the two time frames, and the heterogeneity I do not think this finding warrants as much emphasis as you are putting on it. Four of the six post 2015 studies were in either 2015 or 2016, and seven of the eight pre 2015 studies were from 2013 or 2014. So basically what your findings are saying is that this change occurred between 2013 and 2016, which is both unrealistic, and not how you are presenting the findings. Additionally this is just based on publication date, so to do this properly you would want to look at when the strokes were actually occurring.

Discussion

You start the discussion by saying that this is a systematic review and meta-analysis of 14 studies, but only 12 studies are included in the meta-analysis.

In the results you mention that smaller studies tended to have greater prevalences. There is a fair bit of literature on how smaller size studies tend to overestimate effect sizes. You may want to mention this in the discussion.

I would expect to see some discussion about how your results compare to the results in the GBD-you can get estimates for 15-45 by individual country or region and by year, so it would be interesting to see how similar or different your results are.

You may want to expand your limitations section and be a bit more specific about sources of heterogeneity. There are different age ranges, sample sizes etc.

The final paragraph of your discussion and as well as much of your conclusion focusses on the rising prevalence of stroke, but given the small sample size and very large heterogeneity, I am not sure you can make this strong of a statement based on your results.

**Do you want your identity to be public for this peer review?** For information about this choice, including consent withdrawal, please see our Privacy Policy

Reviewer #1: **Yes: ** Muhammad Luthfi Adnan, MD

Reviewer #2: No

Reviewer #3: **Yes: ** Alma Adler

---

## [Decision Letter · Decision Letter 1]

15 Sep 2025

Prevalence of Stroke in Young Adults in the Middle East and North Africa Region: A Systematic Review and Meta-Analysis

PGPH-D-25-01063R1

Dear Dr. Chivese,

We are pleased to inform you that your manuscript 'Prevalence of Stroke in Young Adults in the Middle East and North Africa Region: A Systematic Review and Meta-Analysis' has been provisionally accepted for publication in PLOS Global Public Health.

Best regards,

Feten Fekih-Romdhane

Academic Editor

Reviewer #1:

Reviewer #2:

Reviewer Comments (if any, and for reference):

Reviewer's Responses to Questions

**Comments to the Author**

Reviewer #1: All comments have been addressed

Reviewer #2: All comments have been addressed

publication criteria?

Reviewer #1: Yes

Reviewer #2: Yes

3. Has the statistical analysis been performed appropriately and rigorously?

Reviewer #1: Yes

Reviewer #2: I don't know

4. Have the authors made all data underlying the findings in their manuscript fully available (please refer to the Data Availability Statement at the start of the manuscript PDF file)?

Reviewer #1: Yes

Reviewer #2: Yes

5. Is the manuscript presented in an intelligible fashion and written in standard English?

Reviewer #1: Yes

Reviewer #2: Yes

Reviewer #1: Some revisions are necessary, such as adjusting the font to a uniform one, as some sections differ. However, overall, the manuscript does not require further revision.

Reviewer #2: The authors had addressed the comments forwarded to them in the previous review. Hence, I recommend acceptance of the manuscript.

**Do you want your identity to be public for this peer review?** For information about this choice, including consent withdrawal, please see our Privacy Policy

Reviewer #1: No

Reviewer #2: No
